# Effectiveness of a Problem-Based Geropsychiatric Nursing Clinical Internship Program

**DOI:** 10.3390/ijerph19074318

**Published:** 2022-04-04

**Authors:** Chia-Shan Wu, Su-Hsien Chang, Kuan-Chia Lin, Jiin-Ru Rong

**Affiliations:** 1Department of Nursing, National Tainan Junior College of Nursing, Tainan 700, Taiwan; shingyeh2001@yahoo.com.tw; 2Department of Senior Citizen Services, National Tainan Junior College of Nursing, Tainan 700, Taiwan; suhsian@yahoo.com; 3Community Research Center, Institute of Hospital and Health Care Administration, National Yang Ming Chiao Tung University, Taipei 112, Taiwan; kuanchia@nycu.edu.tw; 4School of Nursing, National Taipei University of Nursing and Health Sciences, Taipei 112, Taiwan

**Keywords:** geropsychiatric mental health nursing, problem-based learning, clinical internship

## Abstract

Clinical internships that effectively incorporate the care of older adults with mental health disorders are sparse in many countries, including Taiwan. This study investigated the effectiveness of a problem-based geropsychiatric clinical internship program for nursing students in Taiwan. We conducted a quasi-experimental study among 126 nursing students. Experimental and control groups received problem-based geropsychiatric and general psychiatric practice sessions, respectively. Knowledge, attitude, skills, and self-reflection were evaluated before (T1) and after (T2) measurements. There were no significant differences between the groups in knowledge, attitude, skills, and self-reflection at T1. At T2, knowledge was significantly higher in the experimental group (t = 2.39, *p* = 0.02). Attitude, skills, and self-reflection ability did not differ between the groups at T2. Our results showed that clinical problem-based approaches can be applied in geropsychiatric mental health nursing internship programs. The problem-based approach was helpful in improving nursing students’ knowledge about psychiatric symptoms and the health problems of older adults with mental illness. However, it did not significantly enhance or change the attitudes, skills, or the ability to self-reflect among students.

## 1. Introduction

A substantial number of older adults tend to suffer from mental health disorders such as depression, dementia, delirium, alcohol addiction, substance abuse, and suicide. About 20–25% of older adults who are hospitalized or living in nursing homes are reported to have mild depression, with 12% reporting severe symptoms [1,2,3]. Older adults who suffer from mental illnesses such as schizophrenia or depressive disorders need more physical, psychological, and sociological care than those who suffer from other diseases.

Clinical internship is a very important part of nursing education; it provides students with opportunities to practice what they have learned in the classroom, helping them to enhance their knowledge and skills to respond to everyday practice situations and to internalize professional attitudes [4,5,6,7,8]. Clinical internship is also one of the important elements that develop nursing students’ positive recognition of nursing. With the increase in the older adult population globally and older adults reporting more and more mental health problems, there is a need for internship programs that incorporate geropsychiatric clinical practice in order to improve the quality of mental health nursing care. 

Older adults with psychological disorders account for approximately 18% to 55%, and those with early-stage organic mental disorders account for 10% to 24%, of psychiatric medical treatment in Taiwan each year [9]. Currently, there is a lack of systematically planned clinical teaching approaches that focus on the care of older adults with mental health problems in the nursing curriculum in Taiwan. It has been indicated that the classroom teaching of geropsychiatric mental health nursing only accounts for 1/18 of the whole mental health nursing program, with 2 to 3 h teaching time only, with the content mainly geared towards nursing older adults with cognitive dysfunction [10]. Such a lack of adequately planned curriculum may limit nursing students’ practical knowledge in providing mental health nursing care for older adults. Nursing courses and continuing professional education received by new nursing staff tend to focus more on the assessment, diagnosis, and treatment of physical diseases, rather than older adults’ mental health problems, which partly contributes to a lack of ability among nursing students or nursing staff to detect the mental health problems of older adults early and to offer appropriate treatment and support for older adults and their families [11,12]. 

Healthcare students often hold stereotypes about the disabilities, weaknesses, high dependency, and the prevalence of acute diseases among older adults in general [13,14,15,16,17]. Such stereotypes can impact the quality of geropsychiatric mental health nursing [18]. In order to respond to the diverse, changing, and complex health care needs and problems of older adults, clinical nursing teaching methods should not only provide opportunities to practice theoretical knowledge in practical situations, but also incorporate student-centered, problem-based learning approaches to develop their ability to learn and self-reflect, responding to the changing health problems and nursing needs. They should also be equipped to learn independently through self-reflection while using empirical information. 

Experts have called upon nursing educators to place students at the center of their learning, enabling them to enhance their ability for independent learning, reflection, and critical thinking [19,20]. It has also been argued that clinical educators should use their up-to-date professional knowledge of clinical situations to promote students’ independent enquiry by stimulating questions about clinical scenarios, sharing, and managing information appropriately, providing meaningful feedback, and correcting wrong information while helping students to develop new skills [21]. 

Problem-based learning (PBL) is a well-established approach in nursing as well as in other healthcare professions that enable students to develop a comprehensive problem perspective within the healthcare environment. PBL is a learner-centered learning process that triggers learners’ enquiry for knowledge and prepares students for clinical practice in complex environments [21,22]. It has been postulated that PBL fits very well within the nursing curriculum with its emphasis on people, the environment, health, and nursing [22]. PBL provides nursing staff with a more comprehensive perspective of the problem in its own situational nursing environment and support students to become independent learners to actively seek new information and skills for problem solving [21]. By guiding students through enquiry along with providing support and feedback around patient-centered clinical nursing learning, teachers and students have continuous reflection and growth during their clinical practice. 

Clinical nursing teaching tend to remain very educator-centered in Taiwan, and the need for more effective problem-based approaches that can enhance students’ independent ability for active learning, reflection, and critical thinking has been indicated. The evidence on the effectiveness of existing problem-based learning approaches also remains limited, which in turn restricts the ability of nursing educators to incorporate such approaches into the curriculum. This study aimed to evaluate the effectiveness of a clinical problem-based learning of geropsychiatric nursing (CPL-GPN) program on students’ nursing knowledge, attitudes, skills, and ability for self-reflection with respect to the care of older adults with mental illness in Taiwan.

## 2. Materials and Methods

We conducted a quasi-experimental study among two consecutive cohorts of students to evaluate the effectiveness of CPL-GPN compared with regular clinical geropsychiatric learning (RCL-GPN).

### 2.1. Intervention 

The intervention, the CPL-GPN program, was developed based on a review of the literature about PBL approaches as well as the key themes involved in clinical interviews and observations. The program consisted of elements focused on enhancing students’ abilities during internship in seven key areas: 1. questioning and inquiry: ability to identify and explore geropsychiatric nursing issues; 2. construction: ability to search for empirical information on specific nursing issues; 3. thinking: ability to distinguish and appropriately describe, explain, and predict changes in problems; 4. co-learning: ability to cooperate, communicate, and learn with team members; 5. dialogue and expression: ability to listen, express, discuss, and jointly establish an action plan; 6. application: ability to discuss, share, and implement the nursing action plan; and 7. feedback: ability to evaluate the application and impact of the actions on patients. The module included the active ongoing mentorship of the students from their clinical instructors, demonstrations of clinical skills, and explicit guidance for students to solve clinical problems. The comparator, the RCL-GPN module, included regular hands-by-hands teaching with the standard protocol respective to assessments, identifying and managing psychiatric symptoms, building relationships, and therapeutic communication.

The intervention setting was the adult ward of a psychiatric hospital in Taiwan where nursing students undertook a nursing internship for four weeks, with a total of 144 h. About 50% of the patients with mental illness in the adult ward were over 50 years old. An internship instructor was assigned to groups of 7–8 students. The CPL-GPN intervention was administered by trained nursing trainee clinical practice instructors. In both clinical internship programs (CPL-GPN and RCL-GPN), the clinical instructors had Master’s degrees in nursing and had more than 10 years of experience in clinical internship teaching in the psychiatric ward.

However, only the CPL-GPN clinical instructors received training in PBL strategies with opportunities to learn common problem-based learning approaches in clinical practice. The researcher met with the instructors every four weeks to discuss their experiences and to offer advice on any issues. The students in the control group received routine RCL-GPN from a regular clinical instructor.

### 2.2. Participants

The participants comprised of two cohorts of nursing students who were undertaking clinical internships in the adult ward of a psychiatric hospital in Taiwan. The criteria for inclusion were as follows: currently undertaking psychiatric nursing internship; 18 years of age or older; willing to provide written consent by completing and signing the consent form; and had experience in providing nursing care to older adults aged over 50 years with mental illness. Sample size was calculated using G power 3.010 software [23]. The following assumptions were made: α = 0.05, power = 0.80, and effect size = 0.5. The power was set to 0.8; t tests (two tail) were used for two sets of sample estimation; and the minimum sample size was estimated to be 60 students for each group.

### 2.3. Research Ethics

Research Ethics Committee approval was obtained from the Institutional Review Board (IRB) (NTU-REC No.: 201901ES 011). Before carrying out the test, the researchers explained the research purpose in order to gain the interviewees’ consent. The researchers also provided a confidentiality commitment, including the anonymity and confidentiality principles of the research content. Respondents had the right to refuse participating at any time of the survey with the score unaffected.

### 2.4. Study Instruments

The study instruments included a demographic questionnaire, geropsychiatric psychological symptoms and health problem nursing knowledge scale (GPN-K) [24], geropsychiatric psychological symptoms and health problem nursing attitude scale (GPN-A) [24], geropsychiatric psychological symptoms and health problem nursing practice scale (GPN-P) [25], and a self-reflection and insight ability scale (SRIS-C) [26]. The demographic questionnaire included sex, age, grade, and past experience of participating in psychiatric nursing courses for older adults. The GPN-K [24] included 25 questions to assess students’ abilities to evaluate the symptoms of older adults’ mental and physical ill health and the knowledge about measures to be taken while nursing, including the drugs for treatment. Every right answer was scored with one point, and a higher total score implied a mastery of higher levels of knowledge. The Confirmatory Factor Analysis (CFA) of GPN-K showed good fitness and validity. GPN-A [24] contained 19 questions with scores based on a 5-point Likert scale ranging from 1–5 points, respectively, for ‘strongly disagree’, ‘disagree’, ‘neutral’, ‘agree’, and ‘strongly agree’. The higher the score was, the more positive the student’s attitude towards older adults with psychiatric illnesses. Eleven experts were invited to check the validity of the scale’s questions, and the content validity was found to be 0.95. GPN-P covers 5 competence areas and 22 competency tasks based on the core competence framework in clinical nursing practice. The content validity was found to be 0.99. The scores on this scale were based on a 6-point Likert scale with scores ranging from 1–6 points, respectively, for ‘very unskilled’, ‘unskilled’, ‘somewhat unskilled’, ‘somewhat proficient’, ‘proficient’, and ‘very proficient’. The higher the score was, the more proficient the competence was in nursing older adults who are psychiatric patients. The Chinese version of SRIS-C [26] was used by teachers to assess the learning process of students. The scale contained 12 questions on two dimensions of learning: self-reflection, consisting of seven questions, and insight, consisting of five questions, with scores based on a 6-point Likert scale with larger scores representing a higher degree of compliance with the learning process. 

### 2.5. Data Collection Process

A total of 126 nursing students from two internship cohorts were recruited and were randomly assigned into the experimental group (N = 64) and control group (N = 62). All participating students, including those in the experimental and control groups, had successfully completed the theoretical modules in psychiatric nursing in their fourth year. All participants had undertaken similar learning modules including basic nursing in the second year; medical, surgical, obstetric, and gynaecologic nursing in the third year; and psychiatric nursing in the fourth year. Some students had also undertaken elective courses in geriatric nursing and long-term nursing. 

In order to avoid the interaction between the experimental group and the control group, two sample clusters were formed from two student cohorts. The first cohort comprised of those undertaking the internship from 26 November 2018 to 1 January 2019 (control group), and the second cohort comprised of students undertaking the internship from 25 February 2019 to 21 June 2019 (experimental group). Each cluster as a whole was randomly assigned to the experimental group or the control group. The experimental group received CPL-GPN, and the control group received RCL-GPN. Data were collected firstly from the control group and subsequently from the experimental group at two time points: at the start of the internship (T1) and subsequently after the four-week internship (T2). Before enrolling the participants, the researcher explained the research purpose, and written consent was obtained. The researcher also provided a confidentiality commitment, detailing the anonymity and confidentiality principles followed by the researchers and the IRB. Participants were informed of their right to withdraw their participation at any point in time during the study. 

### 2.6. Data Analysis 

Data were analyzed using the Statistical Package for Social Sciences (SPSS), Version 23 (IBM Corp., Armonk, NY, USA). Collected data were checked for accuracy and completeness by the lead researcher before data entry. Descriptive statistics were used to examine frequency distributions, percentages, means, and standard deviations. A chi-square test was used to compare the categorical data, and independent sample t-tests and paired t-tests were used to examine and compare the differences between the experimental group and the control group before and after the intervention. The statistical significance was set as *p* = 0.05. We also conducted a regression analysis using the Generalized Estimating Equation Model (GEE) to control for the presence of significant differences in the baseline between the experimental and control groups in some of the variables such as Grade, Taking Psychiatric Course, Taking Long term Nursing courses, and Practice ward.

## 3. Results

### 3.1. Demographic Characteristics

The demographic characteristics of the participants are presented in Table 1. There were no significant differences in key demographic characteristics such as age and gender between the two groups. More than one third (23; 37.1%) of the students in the control group were in the fifth year of their nursing program, whereas all students in the experimental group were in the fourth year of their program. A higher number (54; 87.1%) of participants in the control group were doing their clinical placement in the acute practice ward compared to the experimental group (X^2^ = 20.00, *p* < 0.001). Sixty (93.8%) students in the experimental group had participated in nursing courses for older adults, which was higher than in the control group (X^2^ = 6.86, *p* = 0.009). Forty one (80%) students in the experimental group had participated in long-term nursing courses, which was higher than the in control group (X^2^ = 6.81, *p* = 0.009).

### 3.2. Learning Effectiveness 

Before the internship (T1), there were no significant differences between the experimental and control groups with respect to knowledge, attitude, skills, and self-reflection of the symptoms of psychiatric and health problems of older adults with mental illness (Table 2). The comparison of the changes in scores for the two groups after the intervention (T2) showed that knowledge about the psychiatric symptoms and health problems of older adults patients with mental illness was significantly higher in the experimental group (t = 2.39, *p* = 0.02), indicating the success of the intervention in improving students’ knowledge of symptoms and health problems of older adults with mental illness. However, there were no significant differences between the experimental and control groups with respect to the attitude, skills, and self-reflection following the intervention (T2). 

The comparison of changes in scores within each group before and after the intervention showed increased levels of nursing knowledge, nursing skills, and self-reflection after the internship (T2) compared with before the internship stage (T2) in both experimental and control groups. The nursing knowledge (GPN-K) scores in the experimental group showed significant increases from 14.97 to 16.84 (t = −3.36, *p* = 0.001), whereas the changes in the scores in the control group was non-significant (t = −0.25, *p* = 0.80). A comparison of the nursing attitude (GPN-A) scores within the groups before and after the intervention showed that the scores decreased in both groups, with a larger decrease in the experimental group, indicating a reduction in positive attitude towards older adults with psychiatric symptoms. Among the 19 questions in GPN-A, 4 questions that showed a significant difference between pre- and post-internship were: GPN-A-10—I think the problems of older adult mental patients are caused by themselves; GPN-A-14—unwillingness to change personality is the main reason for older adults to develop mental illness; GPN-A-16—the problematic behaviour of older adult mental patients is unacceptable; and GPN-A-18—I think the mental illness of the older adults cannot be cured, and there is no need for spending too much time and energy on nursing them. 

A comparison of the nursing skills (GPN-P) scores showed that the nursing skills of the experimental and control groups showed significant improvements after the intervention (T2 compared T1), with a higher increase in 68 scores in the experimental group indicating that, regardless of the intervention, the internship helped students to improve their nursing skills. A comparison of the ability for self-reflection (SRIS-C) scores showed that the ability to self-reflect showed increases both in the experimental and control groups after the intervention (T2 compared T1), with a statistically higher increase in the experimental group, indicating that the intervention helped students to improve their ability to self-reflect (t = −0.57, *p* = 0.02). The difference was evident in the following questions: SRIS-C-5—I often spend time doing self-reflection; SRIS-C-6—it is important for me to be able to understand how ideas are generated; SRIS-C-7—I often think about how I feel about things.

Due to the presence of significant differences in the baseline between the experimental and control groups in some of the variables such as Grade, Taking Psychiatric Course, Taking Long term Nursing courses, and Practice ward, we conducted a regression analysis using Generalized Estimating Equation Model (GEE) controlling for these factors. The findings from this analysis are presented in Table 3. The results showed that knowledge about the psychiatric symptoms and health problems of older adults with mental illness was significantly higher in the experimental group. However, there were no significant differences between the experimental and control groups with respect to their attitudes, skills, and self-reflection following the intervention (T2).

## 4. Discussion

This study examined the effects of a CPL-GPN program on students’ nursing knowledge, attitudes, skills, and self-reflection ability in caring for older adults patients with mental illness. A total of 126 nursing students took part in the study, with 64 students completing the intervention. The results showed that the students who took part in the intervention had significantly higher levels of knowledge about the psychiatric symptoms and health problems of the older adults patients with mental illness, suggesting that the problem-based learning approach was successful in improving their knowledge in these areas. The positive effect of problem-based learning approaches that enable students to inquire, question, express, and reflect in clinical practice situations has been reported by other researchers as well [22,27,28]. 

This study found that the intervention did not significantly enhance or change the attitudes, skills, or the ability to self-reflect among the students. The comparison of the scores before and after the intervention showed that the positive attitude towards older adults with psychiatric symptoms decreased following the intervention. Following the completion of the internship, while the students in our study positively felt the need to care for and empathize with older adults with psychiatric symptoms, and believed that the treatment could improve their condition, they felt it was a rather difficult task to care for older adults with mental health problems, and it was not feasible for the patients to solve the problems by themselves. They even tended to develop more negative thoughts about caring for older adults with mental health problems. The study reported that after five weeks of clinical practice in a psychiatric setting, students’ empathy for patients with mental illness improved, but they were still holding stereotypes and negative attitudes towards mental illnesses for which there is no hope for them to recover, although patients were perceived to be less dangerous [29]. This may be due to the fact that the duration of the students’ internship period, which lasted for five days a week for four consecutive weeks, is not adequate to complete their professional ‘transformation’ where they could fully learn the skills, knowledge, values, behaviours, and attitudes needed in their professional role [30,31]. It has been posited that students’ attitudes are closely linked to a three-stage cognitive process that involves attention, understanding, and the acceptance of content, claims, and characteristics of information sources [32]. 

Our findings showed that the nursing skills of the experimental and control groups showed significant improvements following the intervention, with a higher increase in the experimental group, indicating that regardless of the intervention, the internship helped students to improve their nursing skills. Liao (2018) has reported the positive impact of a 12-week, community-based comprehensive nursing professional development program that involved monthly care discussions for mutual learning. Our findings showed that the ability to self-reflect increased both in experimental and control groups after the internship [33]. Self-reflection involves the examination of individual thoughts and actions leading to an improvement in practice, which in turn leads to increased professional competence [34]. Through self-reflection in practice, ethical literacy and service quality can be improved [35]. It can encourage students to develop the ability for continuous critical self-examination, to reflect on and test their views and behaviours at any time. This will enable them to be able to continuously think, discuss, question, and clarify their own views. Our findings showed that the problem-based clinical learning strategies and activities could potentially improve under continuous guidance, enquiry, and feedback.

A major limitation of this study was the use of convenience sampling and the lack of diversity, as participants were recruited from two internship cohorts in one school, and this could limit the wider applicability of the findings. Due to differences in students taking elective courses and limited hospital capacity, some students undertake internships in their fourth year, once they have completed the theoretical module on psychiatric nursing in the fourth year, whereas others wait until they are in the fifth year to be allocated an internship place. As a result, there were significant differences in the baseline between the experimental and control group clusters with respect to some of the significant variables, such as Grade, Taking Psychiatric Course, Taking Long term Nursing courses, and Practice ward. Although the findings from the GEE modelling showed that the differences with respect to these variables had little impact on the observed differences in outcome variables, it may be possible that the differences in the baseline characteristics may have contributed to the observed differences in outcomes. The use of self-reported data might have also affected the validity of the findings. The school’s psychiatry internship program only lasted four weeks, so it is hard to track the long term effect of the CPL-GPN program on students’ learning. Future studies could add qualitative research to explore the impact of CPL-GPN program on this population to increase the validity of the results.

## 5. Conclusions

In this study, the findings showed that the CPL-GPN internship program had a positive impact on improving students’ knowledge about the psychiatric symptoms and health problems of the older adults patients with mental illness. However, the CPL-GPN program did not significantly enhance or change the attitudes, skills, or the ability to self-reflect among the students.

The use of practical cases to guide learners to understand experiences of interacting with older adults could potentially enhance their ability to explore, construct, think, express, and gain knowledge and skills. Future research should examine the effectiveness of problem-based learning approaches in more depth using rigorous experimental designs involving higher sample sizes across different internship programs. Nested qualitative studies within well-designed trials can be of great value in understanding the experiences of students around problem-based approaches, including the facilitators and barriers in integrating such approaches in clinical internships more generally and in particular in geropsychiatric nursing.

## Figures and Tables

**Table 1 ijerph-19-04318-t001:** Demographic characteristics of the participants (N = 126).

	Control Group (N = 62)		Experimental Group (N = 64)		
	M	SD	M	SD	*p*
Age	19.1	0.07	18.6	0.07	<0.001
	N	%	N	%	*p*
Sex					0.567
Male	3	4.8	4	6.3	
Female	59	95.2	60	93.8	
Grade					<0.001
Fourth year	39	62.9	64	100	
Fifth year	23	37.1	0	0	
Taking Geropsychiatric Nursing courses					0.009
No	14	22.6	4	6.3	
Yes	48	77.4	60	93.8	
Taking Long term Nursing courses					0.009
No	21	33.9	9	14.1	
Yes	41	66.1	55	85.9	<0.001
Practice ward					
Acute ward	54	87.1	32	50	
Chronic Ward	8	12.9	32	50	

**Table 2 ijerph-19-04318-t002:** Comparison of learning effectiveness before (T1) and after (T2) internship (N = 126).

Variable	T1		T2		T2–T1	
	Mean	SD	Mean	SD	t	*p*
Nursing knowledge						
Experimental group	14.97	3.80	16.84	3.47	−3.36	0.001 **
Control group	15.24	2.59	15.37	3.47	−0.25	0.80
t (*p*)	−0.47	(0.64)	2.39	(0.02 *)		
Nursing attitude						
Experimental group	67.81	7.83	66.13	9.63	1.23	0.23
Control group	68.73	6.61	68.71	7.65	0.02	0.99
t (*p*)	−0.71	(0.48)	−1.67	(0.10)		
Nursing skills						
Experimental group	420.59	63.01	488.59	62.32	−7.4	<0.001 ***
Control group	412.98	95.61	486.52	79.03	−5.44	<0.001 ***
t (*p*)	0.53	(0.60)	0.16	(0.87)		
Self-reflection						
Experimental group	49.31	7.51	52.38	8.89	−0.57	0.02 *
Control group	50.13	9.44	51.89	10.66	−1.14	0.26
t (*p*)	−0.54	(0.59)	0.28	(0.78)		

Note: * *p* < 0.05, ** *p* < 0.01, *** *p* < 0.001.

**Table 3 ijerph-19-04318-t003:** GEE analysis of changes in nursing knowledge, nursing attitude, nursing skills, and self-reflection between experimental and control group.

Variables	B	SE	95% CI	*p* Value
Nursing knowledge				
Intercept	14.24	0.71	12.85/15.62	<0.001
Group (EG vs. CG)	−1.748	0.68	−3.20/−0.30	0.018
Time (T2 vs. T1)	1.875	0.55	−2.96/−0.79	0.001
Time × Group	−1.746	0.75		0.020
Nursing attitude				
Intercept	71.57	6.28	59.26/83.87	<0.001
Group (EG vs. CG)	2.29	1.71	−1.07/5.65	0.18
Time (T2 vs. T1)	1.69	1.37	−0.99/4.37	0.22
Time × Group	−1.67	1.67	−4.95/1.61	0.32
Nursing skills				
Intercept	417.34	66.92	286.18/548.49	<0.001
Group (EG vs. CG)	−27.46	16.34	−59.48/4.56	0.93
Time (T2 vs. T1)	−68.00	9.13	−85.90/−50.10	<0.001
Time × Group	−5.53	16.22	−37.32/26.26	0.73
Self-reflection				
Intercept	68.47	7.42	53.92/83.01	<0.001
Group (EG vs. CG)	2.24	2.04	−1.77/6.24	0.27
Time (T2 vs. T1)	−3.06	1.24	−5.49/−0.64	0.01
Time × Group	1.30	1.97	−2.55/5.16	0.51

Notes: Generalized Estimating Equation Model (GEE) were controlled for baseline covariates, grade, taking psychiatric course, taking long term nursing courses, and practice ward.

## Data Availability

The datasets will be available on reasonable request from the corresponding author.

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
