# Peer review of "Effectiveness of a Problem-Based Geropsychiatric Nursing Clinical Internship Program"

_ijerph, 2022, doi:10.3390/ijerph19074318_

Round 1

Reviewer 1 Report

Thanks for inviting me to review the manuscript on the effectiveness of a problem based geropsychiatric nursing clinical internship program. The topic is quite important in mental health. However, there are some concerns needed to be addressed before it can be accepted for publication:

  1. Background: The introduction could be presented in a more concise way.
  2. Study design: More justification for the study design (cluster RCT) is needed. The details for the random allocation algorithm, in particular, when the sample size at the cluster level is 1. Sample size determining is missing in which the ICC across clusters should be taken into account.
  3. Data analysis: There are significant differences in baseline between the two groups, and some of the significant variables like Grade, Taking Psychiatric Course, Taking Long term Nursing courses, and Practice ward, are expected to have influence on the outcome variables, should be controlled when examining the effect of the intervention. It is possible that the observed significant findings in the outcome variables may be due to the differences in these baseline characteristics. In addition, GEE or linear mixed effect models is recommended for analyzing repeated measures. Finally, if it is a cluster RCT, clustering effect should also be controlled.
  4. The results and discussion sections should be revised after revision in data analysis.

Author Response

Dear reviewer:

Thank you for your helpful comments. The manuscript has been revised according to the suggestions.

Title: Effectiveness of a problem based geropsychiatric nursing clinical internship program (Manuscript No: ijerph-1625386).

Point 1:

Background: The introduction could be presented in a more concise way.   

Response 1:We have revised the introduction section to make more concise as suggested.

Point 2: Study design: More justification for the study design (cluster RCT) is needed. The details for the random allocation algorithm, in particular, when the sample size at the cluster level is 1. Sample size determining is missing in which the ICC across clusters should be taken into account.

Response 2: We have now modified the study design as a quasi-experimental design. We have included details of sample size calculation on pages 3 and 4.

We have included the details of the Linear Mixed Models that were used to control for any cluster effects on page 5.

We have also included the findings from this analysis on page 6.

Point 3: Data analysis: There are significant differences in baseline between the two groups, and some of the significant variables like Grade, Taking Psychiatric Course, Taking Long term Nursing courses, and Practice ward, are expected to have influence on the outcome variables, should be controlled when examining the effect of the intervention. It is possible that the observed significant findings in the outcome variables may be due to the differences in these baseline characteristics. In addition, GEE or linear mixed effect models is recommended for analyzing repeated measures. Finally, if it is a cluster RCT, clustering effect should also be controlled.   

Response 3: Thanks for your suggestion. We have now included an additional table (Table 3) with findings from the Generalized Estimating Equation Model (GEE) that was used for regression analysis and verification on page 7. 

We have also included the details of the Linear Mixed Models to control for any cluster effects on page 5.

Point 4: The results and discussion sections should be revised after revision in data analysis.

Response 4: We have now revised the results and discussion sections following the findings from the data analysis.

Reviewer 2 Report

IJERPH-1625386 presents results for a problem-based program in nursing students. While some parts of this paper were interesting, other areas could be improved. I hope the authors consider my feedback.

  • Line 33 and throughout: Avoid using “elderly” and instead consider “older adults” in scientific papers.
  • Lines 36-37: No need for such precision in decimals if “approximately” is being used.
  • Lines 58-63: One sentence paragraph?
  • Results: No need to list X2 because it related to p-values. Perhaps list % difference?
  • Table 1 is a bit challenging to read in current format. The same can be suggested with the t-values and mean+-SD differences.
  • Discussion: A limitations paragraph needs to be included.
  • Make any changes to the abstract that align with those made in the text.

Author Response

Dear reviewer:

Thank you for your helpful comments. The manuscript has been revised according to the suggestions.

Title: Effectiveness of a problem based geropsychiatric nursing clinical internship program (Manuscript No: ijerph-1625386).

Point 1: Line 33 and throughout: Avoid using “elderly” and instead consider “older adults” in scientific papers.

Response 1: We have now replaced the term ‘elderly’ with ‘older adults’ throughout the manuscript. 

Point 2: Lines 36-37: No need for such precision in decimals if “approximately” is being used.

Response 2: We have now revised the section as advised.

Point 3: Lines 58-63: One sentence paragraph?

Response 3: We have now revised the section to incorporate the sentence into the previous paragraph.

Point 4: Results: No need to list X2 because it related to p-values. Perhaps list % difference?

Response 4: Thanks for your suggestion. We have now revised Table1 accordingly.

Point 5: Table 1 is a bit challenging to read in current format. The same can be suggested with the t-values and mean+-SD differences.

Response 5: We have now revised Table 1 to enhance the readability.

Point 6: Discussion: A limitations paragraph needs to be included.

Response 6: Thanks for your suggestion. We have now added a paragraph on the limitations on page 9.

Point 7: Make any changes to the abstract that align with those made in the text. Response 7: We have now revised the abstract to reflect the changes in the text.

Round 2

Reviewer 1 Report

pls see the attachment.

Author Response

Dear reviewer:

Thank you for your helpful comments. The manuscript has been revised according to the suggestions.

Title: Effectiveness of a problem based geropsychiatric nursing clinical internship program (Manuscript No: ijerph-1625386). 

1.ICC calculation: It is unclear the ICCs are between cluster or within cluster ICC. Since it is not a cluster RCT, the calculation of ICC is not expected. Suggest delete the related material in the whole manuscript.    

Response: Thanks for your suggestion. We have revised to make as suggested.

2.Line 232-236: It is unclear whether the p-value of 0.02 is for the comparison of the scores in nursing knowledge at T2 between the two groups or for the comparison of the CHANGES in scores from T1 to T2 between the two groups. The presentation of data in Table 2 usually refers to the former.

Response: The comparison of the scores in nursing knowledge at T2 between the two groups.

Thank you for your suggestion. Table 2 has been revised on page 6.

3.Lines 253: the two percentages ‘59.88%’ and ‘67.36%’ regarding nursing knowledge in the experimental group are not clear. Please show it clearly how they were computed from as in Table 2, only scores are presented. It is better to show the results in table.

Response: Thank you for your suggestion.

This paragraph and Table2 has been revised on page 6-7.

4.Lines 255 – 275: No statistical results were presented to support there are significant changes in the outcomes within the groups. Also, if the results at item levels are of interested, it is necessary to present the results for ALL the items in the scale in a table, rather than selecting the significant ones for reporting – It will mask the whole picture. In addition, the results at item level were not discussed in the discussion section. Based on this observation, I suggest no to report results at item level.

Response: Thank you for your suggestion. This paragraph has been revised. We have  add the outcomes within the groups on Table 2.

5.Table 3: indicate which covariates were controlled in the analysis. 

Response: Thanks for your suggestion. We additional the notes of table 3 on page 8. 

6.Line 338: Please provide reference for ‘consecutive cluster sampling’ as it is not a common sampling method if there is such a sampling method. Cluster sampling is a probability sampling. If the clusters are recruited consecutive, then, it cannot be called a cluster sampling as clusters have to be selected randomly from a list of clusters in cluster sampling. Indeed, there is no sampling in the study as all the students in the two cohorts were asked to participate. Another limitation is it is only a quasi-experimental design and hence the results are subject to bias (in particular, selection bias in this situation). Because of the potential selection bias, the authors should not overstate the study findings by using the words like ‘demonstrated the effectiveness ...’

Response: Thanks for your suggestion. We have revised to make as suggested on page 9.